# COVID-19 Vaccine Hesitancy among Parents of Children Younger than 12 Years: Experience from a Tertiary Outpatient Clinic

**DOI:** 10.3390/pharmacy12030085

**Published:** 2024-05-31

**Authors:** Moataz Mohamed Hassan, Laila Al Yazidi, Nagi Elsidig, Mohamed Al Falahi, Najah Salmi, Yahya Al-Jaffari, Labiba Al-Amri, Huyam Zeiidan, Ibrahim Al-Zakwani

**Affiliations:** 1Department of Pharmacy, Sultan Qaboos University Hospital, University Medical City, P.O. Box 35, Alkoudh, Muscat 123, Oman; alfalahi@squ.edu.om (M.A.F.); najahalsalmi@squ.edu.om (N.S.); yahyaj@squ.edu.om (Y.A.-J.); l.alaamri1@squ.edu.om (L.A.-A.); azakwani@squ.edu.om (I.A.-Z.); 2Department of Child Health, Sultan Qaboos University Hospital, University Medical City, P.O. Box 35, Alkoudh, Muscat 123, Oman; lailay@squ.edu.om (L.A.Y.); nagiamin@squ.edu.om (N.E.); 3Emergency Medical Department, Sultan Qaboos University Hospital, University Medical City, P.O. Box 35, Alkoudh, Muscat 123, Oman; h.elmutwakil@squ.edu.om; 4Department of Pharmacology and Clinical Pharmacy, College of Medicine & Health Sciences, Sultan Qaboos University, P.O. Box 35, Alkoudh, Muscat 123, Oman

**Keywords:** vaccine hesitancy, children, COVID-19, SARS-CoV-2, Oman

## Abstract

This study explored parents’ attitudes towards vaccinating their children against COVID-19 and the rate of vaccine hesitancy in Oman. A cross-sectional, online, self-administered questionnaire, previously validated and administered between June 2021 and May 2022, was used. The questionnaire consisted of nine items. Parents of children younger than 12 years were eligible for participation. A total of 384 participants, including 207 males (54%), completed the questionnaire, resulting in an 86% response rate (384/447). The results showed that 69% of participants were hesitant to vaccinate their children aged 1–11 years. In parents of children aged 1–4 years, vaccination status was significantly associated with vaccine hesitancy (odds ratio [OR], 0.116; 95% confidence interval [CI], 0.044–0.306; *p* = 0.001). Furthermore, after multivariable analysis, compared to the fathers, mothers were significantly less likely to be associated with vaccine hesitancy (OR, 0.451; 95% CI, 0.240–0.848; *p* = 0.013).

## 1. Introduction

Since 2019, vaccine hesitancy (VH) has been among the top 10 global health threats [1]. VH, as defined by the World Health Organization (WHO) Strategic Advisory Group of Experts (SAGE), refers to a delay in both accepting and declining immunization services despite their accessibility [2]. Vaccination reduces infectious disease incidence, morbidity, and fatality and increases life expectancy [3]. As of May 5, 2024, more than 775 million coronavirus disease 2019 (COVID-19) cases were confirmed globally, including over 7 million deaths [4]. Vaccination against COVID-19 is essential because preventive measures alone are insufficient to inhibit its spread [5]. Globally, COVID-19 affects 1–5% of children aged <18 years [6]. Children with COVID-19 show a wide range of clinical symptoms, from no symptoms to severe infections such as severe pneumonia and multisystem inflammatory disease (MIS-C) [6,7,8]. Fortunately, only a small percentage of children with COVID-19 require extended hospitalization or intensive care, and fatal outcomes are uncommon [9]. Children with chronic diseases (CD) are at a significantly higher risk of hospitalization and severe illness due to COVID-19 than their peers without CD. Children with CD are almost eight times more likely to be hospitalized for COVID-19 and three times more likely to experience severe illness once hospitalized [10]. To date, there have been no reports of COVID-19-related mortalities in children in Oman [8]. According to recent surveys, 60–80% of adults are willing to receive the COVID-19 vaccine, 10% remain unwilling, and the rest are uncertain [11,12,13,14,15]. On October 12, 2022, the Food and Drug Administration (FDA) granted Emergency Use Authorizations (EUAs) for bivalent mRNA vaccines against COVID-19. Pfizer-BioNTech and Moderna developed these vaccines, which were authorized by the FDA for use as a single booster dose at least two months after completing the primary series or monovalent booster vaccination for children aged 5–11 and 6–17 years old, respectively. The FDA amended the EUAs on 8 December 2022, to include children aged six months and older [16,17].

In a systematic review conducted by Galanis et al. (2022), parents’ intention to vaccinate their children against COVID-19 was moderate (60.1%), with studies reporting figures ranging from 25.6% to 92.2% [18]. According to the same systematic review and meta-analysis, factors such as having a father, being older, having a higher income, and perceiving a higher threat from COVID-19 were the primary factors of parents’ intentions to vaccinate their children [18]. Another cross-sectional study assessed parents’ willingness to vaccinate their children against COVID-19 and found that 61.4% of parents were concerned about vaccine safety issues and 54% believed that there was a lack of evidence regarding the safety of new vaccines [19].

Vaccine-hesitant parents have received considerable attention in recent years. Although research has been conducted on parental VH, few studies have examined parental VH in children with CD [20,21,22,23,24,25,26,27,28,29,30,31,32,33,34,35]. Regarding parental COVID-19 VH, infection in CD ranges from 26% to 81% [28,31,36,37,38].

A systematic review conducted by Yam and Huhmann (2022) emphasized that improving accessibility, confidence, and complacency for COVID-19 boosters should be a top priority for public health initiatives [39]. Another systematic review by Galanis et al. (2022) highlighted the need for policymakers to understand parents’ reluctance towards COVID-19 [18]. Alalawi et al.’s (2024) systematic review identified key factors that influence parents’ willingness to vaccinate their children, including building trust in COVID-19 vaccines and increasing vaccination rates [40]. Additionally, regulatory bodies should focus on raising awareness about the safety and efficacy of vaccines, specifically targeting healthcare workers, to positively impact public attitudes towards vaccines and prepare for future pandemics. In Oman, a knowledge gap has been indicated concerning parents’ knowledge, behavior, and hesitation regarding vaccinations for children under 11 years of age with underlying chronic medical conditions. Considering the current epidemiological situation, the present study is crucial in addressing the existing gap in the literature. The aim of this cross-sectional survey was to evaluate the attitudes of parents in Oman with at least one child aged 1–11 years and who had a CD towards COVID-19 vaccination.

## 2. Materials and Methods

### 2.1. Study Design and Settings

This cross-sectional study used electronic data collected through an online self-administered, well-designed, pre-validated Google questionnaire [41]. The investigators, who were native Arabic speakers, translated the questionnaire into Arabic. An expert translated the Arabic version of the questionnaire into English to evaluate the preliminary translation. To test language equivalence, a multilingual expert answered both versions. Under the supervision of the researchers, the participants completed the final Arabic version of the questionnaire, which took approximately eight minutes to complete. A waiver of consent was obtained to indicate the need for participation. Participation in the survey was voluntary, and no compensation was provided. Data were collected from participants between 25 October 2021 and 1 May 2022, based on the following inclusion criteria: parents were required to be 18 years of age or older, and all parents with children under the age of 11 were asked to participate in the survey, attend an outpatient pediatric department, and be able to understand the Arabic language. Questionnaires were administered to one parent (father or mother) of each family member. Parents of children 12 years of age and older and those who did not consent to participate in the survey were excluded.

### 2.2. Vaccine Hesitancy Rate Measurement

The parental questionnaire consisted of sociodemographic information and the VH scale for COVID-19 [41]. The sociodemographic portion included information on the children (date of birth, gender, and medical background) and parents (relationship with the child, date of birth, and educational level). Education levels were divided into five groups: primary school, secondary school, bachelor’s degree, master’s degree, and doctorate. A waiver of consent was obtained to indicate the need for participation.

The “attitudes toward COVID-19 vaccines” section consists of nine statements with a five-point Likert scale (3 = Unsure, 1 = strongly disagree, 2 = disagree, 4 = agree, 5 = strongly agree), with questions about hesitancy and concerns regarding COVID-19 vaccines [41]. The responses to the positively worded questions (1, 2, 3, and 9) were as follows: strongly disagree, 1 point; disagree, 2 points; neither agree nor disagree, 3 points; agree, 4 points; strongly agree, 5 points. The responses to the other questions (4, 5, 6, 7, and 8) were flipped because the questions were negatively worded: strongly agree, 1 point; agree, 2 points; neither agree nor disagree, 3 points; disagree, 4 points; and strongly disagree, 5 points. The total score for all nine items was then summed to a maximum of 45. Lower scores represented higher VH values, and a cut-off value of ≤30 was used to define VH.

### 2.3. Sample Size

The calculated the sample size, assuming that the predicted COVID-19 vaccine coverage was 50% among children, with a margin of error of 5% and a 95% confidence interval, was 384 participants.

### 2.4. Validation of the Arabic Version of the COVID-19 Questionnaire

#### Forward and Backward Translation

Two bilingual translators who spoke Arabic in their native languages translated the tool into Arabic. The translators were health specialists who were familiar with the questionnaire’s contents. Following the translation, two multilingual translators translated the tool back into English. Disagreements between the source and the back-translated versions were also investigated. The bilingual expert panel revised the forward-translated tool as many times as necessary until an agreeable version was created. This study used standard Arabic because it is the official language of 21 Arab nations and is widely taught, understood, and spoken by local Arabs [42].

### 2.5. Internal Reliability

A pilot sample of 30 participants was used to assess the reliability of the questions (test–retest). The items in this section had a Cronbach’s alpha value of 0.91, indicating excellent reliability. The overall Cronbach’s alpha for parents’ hesitancy towards COVID-19 vaccination using VH was 0.81.

### 2.6. Statistical Analysis

Descriptive statistics were used to describe the data. Categorical variables were reported as frequencies and percentages. Differences between groups were analyzed using Pearson’s chi-squared test (Fisher’s exact test was performed for expected cells < five). The mean and standard deviation were used to present the data for continuous variables, and the analysis was performed using the Student’s *t*-test. The impact of demographic and clinical characteristics (age of the child, mother’s care (yes/no), sickle cell disease (SCD) (yes/no), and history of prior infection (yes/no)) on VH (yes/no) was analyzed using multiple logistic regression with the simultaneous method. The goodness-of-fit of the multivariable logistic model was examined using the Hosmer–Lemeshow goodness-of-fit statistic [43]. To measure internal consistency, Cronbach’s alpha was computed for the subscales of the questionnaire. The two-tailed level of significance was set, a priori, at *p* < 0.05. Statistical analyses were performed using STATA version 16.1 (STATA Corporation, College Station, TX, USA).

### 2.7. Ethics

This study was approved by the Medical Research Ethics Committee (SQU-EC/575/2021, MREC #2640; date of approval 7 November 2021). All parents provided written informed consent. De-identified data were collected, and participation was voluntary.

## 3. Results

We approached 447 potential respondents, of whom 63 refused to participate. Therefore, we received 384 caregivers’ data for the analysis, with a response rate of 86%. No data were missing. Table 1 presents the characteristics of the respondents and their children. The survey included both fathers (*n* = 207; 48%) and mothers (*n* = 177; 46%). The mean age of the participants (caregivers) was 35 ± 8 years, while that of the children was 3 ± 0.9 years. An analysis of age distribution among the children demonstrated the highest prevalence in children aged <1 year (*n* = 25; 6.5%), followed by those aged 1–4 years (*n* = 101; 26%), 5–7 years (*n* = 116; 30%), and 8–11 years (*n* = 142; 37%). Approximately 285 (74%) study participants reported a history of CD among their children. Regarding education, 47% (*n* = 181) of the participants had completed secondary school, whereas 41% (*n* = 151) had a BSc degree.

The mean age of the children with CD was 2.6 years, and more than half (56%) of them were female. The most common chronic medical conditions encountered were SCD (*n* = 50; 13% of all cases) and epilepsy (*n* = 32; 8% of all cases) (Table 2). Most children (*n* = 284; 73.90%) had a history of CD, while 100 (26%) were healthy with no history of CD. Parents of children with COVID-19 reported a hesitation rate of approximately 40% (*n* = 106). Children with CD such as depression and systemic lupus erythematosus had parents with high VH rates (100%). Furthermore, parents exhibited a high VH rate when their children were diagnosed with diseases, such as diabetes mellitus (*n* = 8; 89%), SCD (*n* = 36; 72%), and bone marrow or solid organ transplantation (*n* = 21; 70%). There have been few reports on VH. VH was 33% and 38% in parents whose children had cardiovascular disease and thalassemia, respectively. Table 2 shows parents’ responses to each item of the vaccine statement regarding hesitancy. Half of the parents (*n* = 187; 48%) strongly agreed or agreed that a vaccine is an excellent approach to protecting their child from COVID-19. Moreover, almost half of the parents (*n* = 212; 55%) strongly agreed or agreed that their children would have a serious vaccine side effect. Furthermore, almost three-fourths (*n* = 284; 74%) agreed or strongly agreed that they trusted and relied on the information they received regarding COVID-19 vaccinations in children.

Figure 1 shows the sources of information selected by the parents and guardians when deciding whether to vaccinate their children against COVID-19. The majority of parents depended on healthcare organizations (*n* = 219; 57%), WhatsApp (*n* = 185; 48%), or the Internet (*n* = 180; 47%) for COVID-19 vaccination information.

Table 2 lists pediatric diseases and the prevalence of VH and shows that there were no statistically significant differences in VH rates between children with and without CD (25% vs. 27%; *p* = 0.733). Table 3 shows that almost one-third of the parents believed that immunizations were essential for their children. In addition, approximately 62% (*n* = 238) of parents were still determining if they agreed or disagreed with the statement that newer vaccines such as the COVID-19 vaccine had more risks than older vaccines.

Table 4 shows the relationship between the underlying factors and VH in children using multivariable logistic regression. Mothers were less hesitant regarding COVID-19 vaccination than fathers (adjusted odds ratio (aOR) = 0.451, 95% confidence interval (CI) = 0.24–0.848; *p* = 0.013), and parents with children aged 1–4 years were less hesitant to vaccination than those aged <1 year (aOR = 0.116; 95% CI: 0.044–0.306; *p* < 0.001).

## 4. Discussion

To the best of our knowledge, this investigation is the first of its kind to examine a group of Omani parents of children aged 1–11 with chronic health conditions. The primary objective of this survey was to provide insight into their attitudes towards COVID-19 vaccination, specifically their enthusiasm and concerns as well as the factors that influence their beliefs.

To date, considerable information has been published regarding parental COVID-19 VH in individuals below 12 years of age [28,44,45,46,47,48,49,50,51,52,53]. Therefore, data on COVID-19 VH are valuable in guiding the planning of future COVID-19 immunization strategies for children with CD.

Vaccination against COVID-19 is currently the most effective method for minimizing the burden of the COVID-19 pandemic. Our survey at SQUH found that most caregivers were hesitant about vaccinating their children against COVID-19 [48]. In our study on SCD, bone marrow or solid organ transplantation, and blood cancer, parents reported high vaccine hesitancy at 72%, 70%, and 67%, respectively. This is consistent with the results of the previous studies by Elkhadry et al. (2022) [37] and Miraglia et al. (2023) [38]. In contrast, children with cardiovascular disease, thalassemia, and nephrotic syndrome had lower VH rates (33%, 38%, and 48%, respectively). These results were similar to those reported by Wang et al. (2023) [35] and Ma et al. (2022) [31]. The diversity in parental willingness has been confirmed by comparable examinations among the general public, and this variation can be attributed to differences in study designs, study populations, levels of awareness and information, and attitudes towards immunization [18,54,55].

In a systematic review conducted by Chen et al. in 2022, contradictory evidence was found regarding the efficacy of parental and guardian sex as a predictor (OR: 1.1130, 95% CI: 0.5903–2.0986) [19]. Our multivariate analysis revealed that being a mother (aOR, 0.451; 95% CI: 0.240–0.848; *p* = 0.013) was one of the factors influencing vaccine hesitancy, indicating that women were generally less resistant to vaccination than men. Our findings align with those of Babicki and Wan, who reported a higher acceptance rate in their research, similar to the results obtained in our study [56,57]. This could be attributed to the fact that Omani mothers devote more time to their offspring and are more attentive to health issues that affect their children. This insight can serve as a basis for crafting strategies, such as tailoring parental education initiatives to specific genders, for enhancing vaccine acceptance. Notably, we also showed that reluctant parents lacked confidence in vaccination because they were worried about its side effects. According to a systematic review conducted by Iqbal et al., 2023, the primary reason for parents’ decision not to vaccinate their children was the fear of side effects and safety concerns [58].

One possible explanation for the higher overall rate of VH in our study is that younger parents and guardians were aware of the low risk of severe COVID-19 in children. This assumption is consistent with previous studies showing that parents or guardians aged >50 years have greater vaccination confidence [59]. Furthermore, the incidence of VH was much higher in children younger than 11 years, which affected their parents’ perspectives regarding vaccination.

Our study had some limitations. First, this was a single-center study from a tertiary care center where the patients came from across the country. In addition, the case mix and severity of children may not be representative of the country. Moreover, since this was a retrospective cross-sectional study, its design limits its ability to establish a causal-effect relationship. This design could also have led to biases including social desirability bias, self-selection bias, non-response bias, and residual compounding. Therefore, future research should examine parental VH in different locations in Oman. In addition, regular childhood immunizations, such as the influenza vaccine, were not recorded for this age group of children as a control comparison, and the relationship between parents’ and healthcare workers’ VH was also not evaluated. Finally, the strengths of our study include the use of a validated questionnaire to assess VH and obtaining a high response rate [33].

VH can result in multiple public health implications. Unvaccinated individuals create pockets of susceptibility, allowing the virus to continue circulating, potentially leading to outbreaks or the resurgence of COVID-19. High vaccination rates are crucial for achieving community immunity and reducing the overall transmission of the virus [60]. Children who cannot receive the COVID-19 vaccine because of their age or medical conditions depend on community immunity for protection. If VH among other children leads to lower overall vaccination rates, the risk to vulnerable populations, such as immunocompromised individuals or those with underlying diseases who may be at a higher risk of severe illness or complications from COVID-19, increases [59].

Furthermore, VH among children can contribute to school disruptions and outbreaks. If a significant number of children remain unvaccinated, the likelihood of COVID-19 transmission within an educational setting increases. This can result in temporary closures of schools and quarantines and disruptions to in-person learning, affecting the educational experience and overall well-being of children [61].

In addition, vaccinating children is essential to controlling the COVID-19 pandemic. Children can contribute to the spread of the virus even if they are less likely to experience severe illness. VH among children can delay the achievement of widespread vaccination coverage, thereby prolonging the duration of the pandemic and its associated economic, social, and health impacts [62].

## 5. Conclusions

In our study, parents’ hesitancy toward giving their children the COVID-19 vaccine was higher than that reported in previous studies. Additionally, the female sex and the age of the child were factors that increased the risk of VH. Parental VH must be urgently addressed to improve vaccine uptake in guardians of children attending pediatric outpatient departments. Therefore, effective health education and awareness campaigns should focus on this age group to help parents immunize their children against SARS-CoV-2, which could decrease the spread of the virus in asymptomatic individuals and strengthen their immune systems.

## Figures and Tables

**Figure 1 pharmacy-12-00085-f001:**
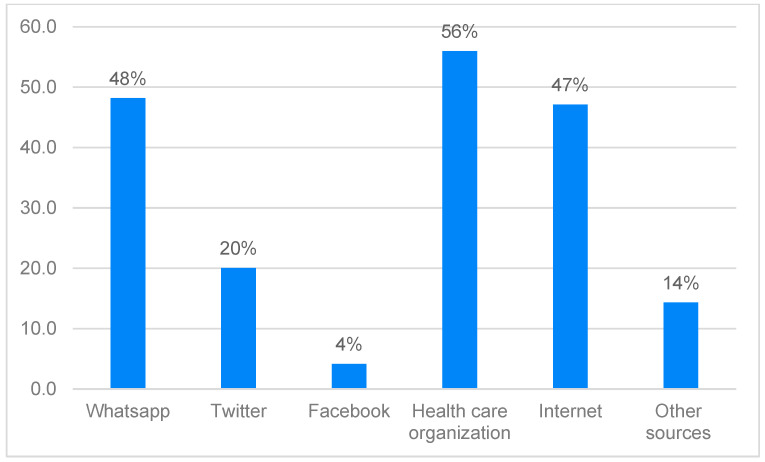
Parents’ sources of information about COVID-19.

**Table 1 pharmacy-12-00085-t001:** Vaccine hesitancy according to participant characteristics (N = 384).

Characteristic	Value	Hesitant *n* (%)	*p* Value
Child			
Age (years), mean ± SD	2.9 ± 0.94		
Gender n (%)			0.912
Male	168 (44)	116 (44)	
Female	216 (56)	148 (56)	
Age group (years), n (%)			0.227
<1	25 (6.5)	20 (80)	
1–4	101 (26.3)	74 (73)	
5–7	116 (30.2)	80 (69)	
8–11	142 (37)	90 (63)	
Relationship of respondent to child			0.283
Father	207 (54)	123 (66)	
Mother	177 (46)	141 (71)	
Age group			0.744
18–24	24 (6)	17 (71)	
25–34	129 (34)	94 (73)	
35–44	192 (50)	126 (66)	
45–54	36 (9)	25 (69)	
≥55	3 (1)	2 (67)	
Educational level of the respondent			0.503
Primary school	15 (4)	13 (87)	
Secondary school	181 (47)	121 (67)	
Bachelor’s degree	159 (41)	110 (69)	
Master’s degree	25 (7)	18 (72)	
Doctoral degree	4 (1)	2 (50)	

**Table 2 pharmacy-12-00085-t002:** Prevalence of vaccine hesitancy according to the child’s disease.

Disease in Child	Hesitant N (%)	Non-Hesitant N (%)	*p* Value
Diabetes mellitus	8 (89)	1 (11)	0.187
Cystic fibrosis	5 (56)	4 (44)	0.387
Sickle cell disease	36 (72)	14 (28)	0.595
Bone marrow or solid organ transplantation	21 (70)	9 (30)	0.878
Genetic/metabolic disorders	11 (64)	6 (35)	0.713
Systemic lupus erythematosus	2 (100)	0 (0)	0.339
Cardiovascular disease	2 (33)	4 (67)	0.059
Rheumatoid arthritis	3 (75)	1 (25)	0.786
Nephrotic syndrome	3 (43)	4 (57)	0.136
Thyroid disorder	7 (87)	1 (13)	0.248
Asthma	14 (64)	8 (36)	0.594
Thalassemia	6 (38)	10 (62)	0.006
Hemophilia	1 (25)	3 (75)	0.058
Blood cancer	20 (67)	10 (33)	0.798
Epilepsy	26 (81)	6 (19)	0.111
Depression	2 (100)	0 (0)	0.339
Attention deficit hyperactivity disorder	6 (75)	2 (25)	0.700
Other chronic disease	17 (59)	12 (41)	0.221
Non-chronic disease	74 (75)	25 (25)	0.135

**Table 3 pharmacy-12-00085-t003:** Parental vaccine hesitancy: nine statements and responses (*N* = 384).

Questions	Strongly Agree	Agree	Strongly Disagree	Disagree	Unsure
(1) Getting the vaccine is an excellent way to protect my child from COVID-19. (P)	55 (14)	132 (34)	28 (7)	48 (13)	121 (32)
(2) Having my child vaccinated is important for the health of others in my community. (P)	94 (24)	144 (38)	14 (4)	42 (11)	90 (23)
(3) The information I receive about the COVID-19 vaccine from my child’s healthcare provider is reliable and trustworthy. (P)	74 (19)	210 (55)	3 (1)	15 (4)	82 (21)
(4) New vaccines like COVID-19 vaccines carry more risks than older vaccines. (N)	35 (9)	71 (19)	9 (2)	31 (8)	238 (62)
(5) I am concerned about the severe side effects of the COVID-19 vaccine. (N)	86 (22)	126 (33)	10 (3)	51 (13)	111 (29)
(6) I think the COVID-19 vaccines might cause short-term problems for my child, like fever, pain at the injection site, and fatigue. (N)	64 (17)	150 (39)	7 (2)	39 (10)	124 (32)
(7) I think the COVID-19 vaccine might cause long-term health problems for my child. (N)	37 (10)	70 (18)	14 (4)	76 (20)	187 (48)
(8) I think my child will not get sick with COVID-19 illness even if they do not get the COVID-19 vaccines. (N)	24 (6)	78 (20)	33 (9)	73 (19)	176 (46)
(9) COVID-19 could make my child very sick. (P)	37 (10)	71 (19)	14 (4)	76 (20)	186 (47)

P, positive statements (1, 2, 3, and 9); N, negative statements (4, 5, 6, 7, and 8).

**Table 4 pharmacy-12-00085-t004:** Multivariable analysis of demographic and clinical characteristics regarding parental vaccine hesitancy.

Characteristic	Odds Ratio (95% CI)	*p* Value
Age group of child (years)		
<1	1.000 (reference)	
1–4	0.116 (0.044–0.306)	0.001
5–7	0.685 (0.312–1.504)	0.346
8–11	0.988 (0.459–2.128)	0.975
Relationship of respondent to the child		
Father	1.000 (reference)	
Mother	0.451 (0.240–0.848)	0.013
Sickle cell disease	0.629 (0.345–1.149)	0.132
Has your child ever been infected?	1.453 (0.752–2.808)	0.266

## Data Availability

Data are available from the corresponding author on reasonable request.

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
