# Peer review of "COVID-19 Vaccine Hesitancy among Parents of Children Younger than 12 Years: Experience from a Tertiary Outpatient Clinic"

_pharmacy, 2024, doi:10.3390/pharmacy12030085_

Round 1
Reviewer 1 Report
Comments and Suggestions for Authors
The manuscript entitled "Questionnaire-Based Investigation of 2 COVID-19 Vaccine Hesitancy among Omani Parents in the 3 Outpatient Pediatric Departments of Sultan Qaboos University 4 Hospital: A Cross-Sectional Study" is a well written article.
Some minor points to consider:
1. Line 22: parenthesis
2. VH or V.H. please remain consistent through the text
Please check again the whole article for word editing
Author Response
Reviewer 1
The manuscript entitled "Questionnaire-Based Investigation of 2 COVID-19 Vaccine Hesitancy among Omani Parents in the 3 Outpatient Pediatric Departments of Sultan Qaboos University 4 Hospital: A Cross-Sectional Study" is a well written article.
Some minor points to consider:
Line 22: parenthesis
Authors’ reply: Thank you for your comment. The parentheses have been modified accordingly.
VH or V.H. please remain consistent through the text.
Authors’ reply: Thank you for your comment. V H has been modified to VH throughout the manuscript.

Reviewer 2 Report
Comments and Suggestions for Authors
I enjoyed reading the paper. It is interesting and useful.
My one concern is that authors report that vaccine hesitancy is higher among mothers than fathers (71% vs 66%). Since, there is no statistically significant difference between mothers and fathers on this indicator, it is incorrect to say that hesitancy is higher among mothers.
This mistake should be corrected.
Author Response
Reviewer 2
My one concern is that authors report that vaccine hesitancy is more likely higher among mothers than fathers (71% vs 66%). Since, there is no statistically significant difference between mothers and fathers on this indicator, it is incorrect to say that hesitancy is higher among mothers.
This mistake should be corrected.
Authors’ reply: Thank you for your comment. You are absolutely correct. This was a typo. Mothers were, in fact, less likely to be associated with vaccine hesitancy when compared to fathers (aOR, 0.451; 95% CI: 0.044-0.306; p = 0.001). This has now been modified through the text.

Reviewer 3 Report
Comments and Suggestions for Authors
This paper is a study of parental hesitancy in giving their children vaccination against novel coronavirus infection. The research topic itself is very interesting.
The parents included in the study were parents of children attending paediatric outpatient clinics. The results show that the children have some underlying disease. How do you think this affected the results of the present study? Also, compared to previous studies, what are the selection results for this parent?
What does it add to the subject area compared with other published material?
Author Response
Reviewer 3
This paper is a study of parental hesitancy in giving their children vaccination against novel coronavirus infection. The research topic itself is very interesting.
The parents included in the study were parents of children attending paediatric outpatient clinics. The results show that the children have some underlying disease. How do you think this affected the results of the present study? Also, compared to previous studies, what are the selection results for this parent?
Authors’ reply: Thank you for your comment. As shown in Table 4, adjusting for other factors in the model, sickle cell disease, which was prevalent in this pediatric population, was not a significant predictor of vaccine hesitancy (aOR, 0.629; 95% CI: 0.345-1.149; p = 0.132).
What does it add to the subject area compared with other published material?
Authors’ reply: The frequent visits to healthcare facilities by children with underlying health conditions heighten their susceptibility to COVID-19, emphasizing the need for vaccination as a means of achieving immunity for these vulnerable individuals. Addressing parents' concerns by their familiar paediatricians about vaccine safety can also boost COVID-19 vaccine uptake.
Authors’ reply (continuation): Recognizing the reasons behind parental hesitancy towards COVID-19 vaccines can aid policymakers in altering preconceived notions and implementing widespread COVID-19 immunization programs. By identifying the elements that impact parents' readiness to vaccinate their children against COVID-19, it is possible to increase parents' confidence in COVID-19 vaccines and ensure that more children receive the vaccine.

Reviewer 4 Report
Comments and Suggestions for Authors
This investigation addresses an important topic. However, Major revision is required to improve this paper further. Please see my comments below, which might help to enrich your research further:
1. Title: The title is a bit long. Consider making it shorter, something similar.
“Investigation of COVID-19 Vaccine Hesitancy among Omani Parents in the 3 Outpatient Pediatric Departments of Sultan Qaboos University Hospital”
2. Abstract:
a) Line 15-20: The method section is unclear, redundant, and does not flow well and should be improved. For example, the following three sentences can be merged.
“This cross-sectional study used an online, self-administered questionnaire. The 9-item questionnaire, validated earlier, was administered between June 2021 and May 2022. The parents received a pre-validated Google questionnaire.”
b) Also, the presentation and readability of the material should be improved. What is “accepted” in this sentence: “Responses from the parents of children younger than 11 years of age 19 were accepted.” Write more explicitly and precisely.
c) Line 20-21: You do not need the response rate in the abstract. Either remove or merge these two sentences:
“A total of 384 participants completed the questionnaire, including 207 males 20 (54%). The response rate was 86% (384/447).”
3. Introduction
a) The introduction section is exceptionally brief. The main flaw is that the original contribution of the present study over past research has not been discussed. The review of relevant studies on the topic is inadequate. Thus, the original contribution of the present study over past research is unclear. Conducting a study in a new country does not necessarily warrant a new study. Thus, the author should discuss how this investigation adds to existing research in more detail. The need/rationale for the study should be discussed in greater detail.
b) The definition of vaccine hesitancy can be helpful.
c) You must cite more recent studies, especially from 2023 and 2024, if any. Also, consider citing the conclusion of systematic reviews on COVID-19 vaccine hesitancy to support your arguments (see below some papers).
Willingness, refusal and influential factors of parents to vaccinate their children against the COVID-19: A systematic review and meta-analysis. Preventive medicine, 157, 106994. https://doi.org/10.1016/j.ypmed.2022.106994
Why Some People Are Hesitant to Receive COVID-19 Boosters: A Systematic Review. Trop. Med. Infect. Dis. 2023, 8, 159. https://doi.org/10.3390/tropicalmed8030159
COVID-19 vaccine hesitancy among healthcare workers in Arab Countries: A systematic review and meta-analysis. Plos one, 19(1), e0296432.
d) Clearly state research objectives and/or questions in line with your key results.
4. Methods.
- There are several subheadings in the Materials and Methods. Some are not critical (e.g., 2.3, 2.9).
- Section 2.2. and 2.4 should be merged.
- Section 2.8 is lengthy. Condense it.
5. Presentation of results
- This is a lengthy section. There are several results (9 tables! and figures). The authors did not provide research questions, and thus, it is difficult to follow and comprehend the results that have been presented in a scattered manner.
- Some results are not critical (e.g., Table 3). Instead, the authors should focus on conclusive results involving tests of significance. Again, the results should be presented in line with and in order of research objectives/questions.
- Line 297-353, too much information; not well-organized and difficult to follow. Results (tables 5-9) about the validity and reliability, including factor analysis, should come before logistic regression results. Some tables (e.g., Tables 5 to 9) are not critical, and thus, they can be condensed/merged or can be included in the appendix
6. Implication: This study offers some important findings. However, the authors should discuss the implications in greater detail, especially exhibiting the public health implications of the findings.
Comments on the Quality of English Language
None
Author Response
Title: The title is a bit long. Consider making it shorter, something similar.
Authors’ reply: Thank you for your excellent feedback. This has been now changed to “COVID-19 Vaccine Hesitancy among Parents of Children Younger Than 12 Years: Experience from a Tertiary Outpatient Clinic”.
Abstract:
- a) Line 15-20: The method section is unclear, redundant, and does not flow well and should be improved. For example, the following three sentences can be merged.
The present study conducted a cross-sectional analysis utilizing a self-administered online questionnaire. The 9-item questionnaire, which had been validated, was distributed to parents via the internet between June 2021 and May 2022.
Authors’ reply: Thank you for your comment. The three sentences were then merged into two.
- b) Also, the presentation and readability of the material should be improved. What is “accepted” in this sentence: “Responses from the parents of children younger than 11 years of age 19 were or the study should be discussed in greater detail.
Authors’ reply: Thank you for your feedback. This issue has been addressed in the abstract section and highlighted.
- C) The definition of vaccine hesitancy can be helpful.
Authors’ reply: Thank you for your comment. The definition of vaccine hesitancy has now been added to the Introduction section, as follows:
Vaccine hesitancy (VH), as defined by the World Health Organization (WHO) Strategic Advisory Group of Experts (SAGE), refers to the decision to either defer or delay vaccination despite having access to vaccination services.
- d) You must cite more recent studies, especially from 2023 and 2024, if any. Also, consider citing the conclusion of systematic reviews on COVID-19 vaccine hesitancy to support your arguments (see below some papers).
COVID-19 vaccine hesitancy among healthcare workers in Arab Countries: A systematic review and meta-analysis. Plos one, 19(1), e0296432.
Willingness, refusal and influential factors of parents to vaccinate their children against the COVID-19: A systematic review and meta-analysis. Preventive medicine, 157, 106994. https://doi.org/10.1016/j.ypmed.2022.106994
Why Some People Are Hesitant to Receive COVID-19 Boosters: A Systematic Review. fTrop. Med. Infect. Dis. 2023, 8, 159. https://doi.org/10.3390/tropicalmed8030159.
Authors’ reply: Thank you for your suggestion to add recent pertinent references. References for 2023 and 2024 have now been cited in the manuscript as follow:
References for 2023: Number 20,22,26,29, and 34.
References for 2024: Number 40 and 54.
Systematic reviews on COVID-19 vaccine hesitancy has been cited to in the last paragraph
References numbers 39 and 40 are cited in lines 65 and 68, respectively.
- e) Clearly state research objectives and/or questions in line with your key results.
Authors’ reply: Thank you for your comment. The aim of the study has now been written in the last paragraph of the Introduction section.
- Methods.
- There are several subheadings in the Materials and Methods. Some are not critical (e.g., 2.3, 2.9).
- Section 2.2. and 2.4 should be merged.
- Section 2.8 is lengthy. Condense it.
Authors’ reply: Thank you for your comment. As suggested, Sections 2.2. and 2.4 are now merged as one section while Section 2.8 has now been condensed.
- Presentation of results
- This is a lengthy section. There are several results (9 tables! and figures). The authors did not provide research questions, and thus, it is difficult to follow and comprehend the results that have been presented in a scattered manner.
- Some results are not critical (e.g., Table 3). Instead, the authors should focus on conclusive results involving tests of significance. Again, the results should be presented in line with and in order of research objectives/questions.
- Line 297-353, too much information; not well-organized and difficult to follow. Results (tables 5-9) about the validity and reliability, including factor analysis, should come before logistic regression results. Some tables (e.g., Tables 5 to 9) are not critical, and thus, they can be condensed/merged or can be included in the appendix.
Authors’ reply: Thank you for the valuable feedback. We have worked on the Results section again to remove unnecessary repeating statements and in line with the study’s objectives. As per your suggestions, we have also reduced the numbers of tables and figure.
- Implication:This study offers some important findings. However, the authors should discuss the implications in greater detail, especially exhibiting the public health implications of the findings.
Authors’ reply: Thank you for the valuable feedback. The public health implications of the findings were added in the Discussion and Conclusion sections. Please see statements that have now been added to this fact.
Additional research on the progression of the pandemic among pediatric patients, the emergence of new variants, and the factors affecting vaccine efficacy can enhance trust in government, health institutions, and experts. This can assist parents in making informed decisions regarding COVID-19 vaccine administration for children aged 1-11. Moreover, the development of more effective vaccines is crucial to increase COVID-19 vaccine acceptability among parents of children aged 1-11. Primary care pediatricians play a vital role in educating and advising parents about vaccines, making them the most important point of reference. These Physicians are well-equipped to recognize vulnerable children and promote vaccination, despite the significance of prevention in healthy youngsters.

Round 2
Reviewer 3 Report
Comments and Suggestions for Authors
I have reviewed the revisions to the paper and am satisfied with it.
Reviewer 4 Report
Comments and Suggestions for Authors
Thanks for incoporating my suggestions. Good luck.